# Exploring the barriers and facilitators to accessing and utilising mental health services in regional, rural, and remote Australia: A scoping review protocol

**Bianca E. Kavanagh**[1,2]*, **Hannah Beks**[2], **Vincent L. Versace**[2], **Shae E. Quirk**[1,3,4], **Lana J. Williams**[1]

**1** Institute for Mental and Physical Health and Clinical Translation, School of Medicine, Barwon Health, Deakin University, Geelong, VIC, Australia, **2** Deakin Rural Health, Deakin University, Geelong, VIC, Australia, **3** Institute of Clinical Medicine, Psychiatry, University of Eastern Finland, Kuopio, Finland, **4** Institute of Clinical Medicine, Kuopio Musculoskeletal Research Unit (KMRU), University of Eastern Finland, Kuopio, Finland

* bianca.kavanagh@deakin.edu.au

**Data Availability Statement:** No datasets were generated or analysed during the current study. All

## Abstract

### Introduction

Australians from regional, rural, and remote areas face diverse and complex challenges in accessing and utilising mental health services. Previous research has pointed to a range of individual, community, structural, and systemic barriers at play, however, limited literature has synthesised the knowledge on this topic. Parallel to this, information on the facilitators to accessing and utilising mental health services for this group is not well documented. This protocol describes the methodology to undertake a scoping review, which aims to explore the barriers and facilitators associated with accessing and utilising mental health services in regional, rural, and remote Australia. In addition, the scoping review aims to geographically map the identified barriers and facilitators.

### Methods

This protocol is guided by Arksey and O'Malley's methodological framework. A search strategy will be developed and implemented to identify relevant peer-reviewed and grey literature. Studies will be included if they report on the barriers and/or facilitators associated with accessing and/or utilising mental health services in regional, rural, and remote Australia. Two reviewers will independently screen the data at the title/abstract and full-text stage. One reviewer will extract the relevant data using a predetermined charting form and a second reviewer will validate the included data. A Geographical Information System program will be used to map the location of the studies; locations will be stratified according to the Modified Monash Model and relationships between barriers and facilitators will be analysed. Key findings will be presented in a narrative account and in text, tables, and maps.

relevant data from this study will be made available upon study completion.

**Funding:** HB and VLV are supported by the Rural Health Multidisciplinary Training program, funded by the Australian Government. LJW is supported by a National Health and Medical Research Council (NHMRC) Emerging Leadership Fellowship [1174060]. The funders had no role in study design, data collection and analysis, decision to publish, or preparation of the manuscript.

**Competing interests:** The authors declare that they have no competing interests.

**Abbreviations:** ASGS-RA, Australian Statistical Geography Standard Remoteness Area; CINAHL, Cumulative Index of Nursing and Allied Health Literature; EMBASE, Excerpta Medica Database; JBI, Joanna Briggs Institute; GIS, Geographical Information System; GP, General practitioner; MMM, Modified Monash Model; PCC, Population/ concept/context; PRESS, Peer Review of Electronic Search Strategies; PRISMA-P, Preferred Reporting Items for Systematic Reviews and Meta-Analysis Protocols; PROSPERO, International Prospective Register of Systematic Reviews; PsycINFO, Psychological Information Database; RHMT, Rural Health Multidisciplinary Training.

## Discussion

This scoping review will provide a contemporary account on the barriers and facilitators to accessing and utilising mental health services for regional, rural, and remote Australians. It is anticipated that the results of this scoping review will have national policy relevance and may be useful to healthcare providers.

## Introduction

The incidence of mental disorders among individuals located in regional, rural, and remote areas of Australia is comparable to their metropolitan counterparts [1]. However, individuals from regional, rural, and remote settings—defined as areas outside of Australian Major Cities [2]—tend to experience poorer mental health-related outcomes. Regional, rural, and remote Australia is a heterogenous area which is comprised of a variety of coastal, agricultural, and mining communities, with varying levels of population density, often with vast distances between communities and services [3]. In part, these factors contribute to inequalities in access to mental health services. In addition, mental disorder burden appears to increase with the degree of remoteness, suggesting that those in the most remote locations experience the poorest mental health outcomes. This is demonstrated by a two-fold increased rate of mental health-related emergency presentations [4] and deaths by suicide in Very Remote compared to Major Cities of Australia [5].

There is growing literature to suggest that the inequalities in accessing mental health services may be due to specific individual as well as community factors. Access to healthcare is proposed to be a measure of the alignment between healthcare provider/services and the characteristics/expectations of clients [6]; this concept involves aspatial (i.e., social) or spatial (i.e., geographical) dimensions [6, 7]. Aspatial dimensions include *affordability*, *acceptability*, and *accommodation*, relating to healthcare cost, cultural considerations, and communication effectiveness, respectively [6, 7]. Aspatial factors affecting access to healthcare in regional, rural, and remote Australia include lack of help-seeking behaviours, confidentiality concerns, stoicism [8], and social matters [9]. Spatial dimensions involve *availability*—associated with the capacity to deliver the service, and *accessibility*—linked to the cost of travel between the provider and patient [6, 7]. These factors involve geographical isolation [9], and concerns regarding multiple-roles [9], workforce shortages [10] and constrained resources [10] for the healthcare provider. Utilisation of healthcare services involves the development of a healthcare plan during a healthcare encounter and its ensuing implementation [11]. Access and utilisation are separate and related concepts, with utilisation presuming access [11]. Thus, difficulties in accessing healthcare services are likely to affect service utilisation, client satisfaction, and provider practice patterns [6].

In Australia approximately 28% of the population live in regional, rural, and remote areas [12]. The Australian Government recognises that people residing in these areas are considered to have difficulty obtaining medical assistance and that access to healthcare may be delayed and more expensive [13]. In part, this may be due to the increasingly unequal distribution of the health workforce relative to the population in Major Cities compared to other remoteness areas in the last decade. In 2020, there was a greater number of clinical full-time equivalent mental health specialists per/100,000 population working in Major Cities compared to other remoteness areas (i.e., 14.8 vs 2.6–6.5 psychiatrists, 97.0 vs 31.5–83.8 mental health nurses, 95.9 vs 23.5–59.9 psychologists) [14]. Workforce maldistribution of health professionals in

rural Australia may compound health inequalities [15–18] and this is addressed directly at a national scale by the Rural Health Multidisciplinary Training (RHMT) program [19], with the program intent to maximise investment in Inner Regional to Very Remote areas of Australia. Recent data has shown that across most Australian jurisdictions, individuals who live in Metropolitan Areas tend to experience greater relative socioeconomic advantage—highlighting the critical importance of targeted investment in rural areas through programs such as the RHMT [3].

Recently, the Royal Commission into Victoria's Mental Health System (the Royal Commission) recommended incentives to attract and retain the mental healthcare workforce in rural and regional areas [20]. It is likely that similar strategies to reduce barriers and increase facilitators to accessing mental health services may also be adopted in other Australian states and territories, assisting with workforce maldistribution and health inequalities. Further, the Commission highlighted that the impact of climate emergencies will have lasting impacts on the wellbeing, businesses, and livelihoods of rural and regional communities [20]. The Commission emphasised the need to embed digital service delivery within the healthcare system to reduce barriers to access services for rural and regional individuals [20]. Telehealth has previously been shown to be effective in providing 24-hour/day access to timely and expert care—affecting clinical decision making and reducing the number of patients being admitted into mental health inpatient units located outside of their local communities [21]. Telehealth appointments for mental health have had considerable uptake since the beginning of the COVID-19 pandemic, coinciding with the temporary inclusion of telehealth items on the Medicare Benefits Schedule—reported to be welcomed among rural health patients and providers [22]. Thus, telehealth may be a facilitator to access mental health services for rural Australians going forward.

Research that documents inequalities in accessing and utilising mental health services may moderate the disease burden experienced by regional, rural, and remote populations and assist in equitable resource allocation [23]. Spatial distribution may be a useful tool in assessing where the barriers and facilitators to accessing and utilising mental health services exist across Australia, particularly considering that the distance and spatial availability of mental health services reduce their utilisation [24–26]. Spatial distribution permits visualisation and may help to identify priority areas of where barriers need to be overcome and provide examples of facilitators that may be implemented in other areas. Spatial distribution may therefore aid in policy making, especially in relation to resource allocation, workforce shortages, and postulation of service utilisation patterns [26, 27].

Literature synthesising the barriers and facilitators to accessing and utilising mental health services across rural and regional Australia is scarce. One parliamentary inquiry found that despite the social characteristics of rural and remote communities that support positive mental health (i.e., resilience and a sense of community), several factors that are bound to location (i.e., geographical isolation, environmental adversity, lower socioeconomic circumstances, and restricted access to services) place individuals at risk of poor mental health [28]. Further, for some specific communities, such as Aboriginal and Torres Strait Islander peoples, these risk factors are heightened by the historic and cultural experiences of socioeconomic disadvantage and inter-generational trauma [28]. Aboriginal and Torres Strait Islander peoples represent a considerably larger proportion of the remote and very remote Australian population (12% compared to 1% of non-Indigenous people) [29]. Other communities, such as farming and mining communities, are also predominantly located in rural and remote areas of Australia. Although these population groups may feature in the literature, the proposed scoping review will not centre on specific population groups of interest, rather a broad approach to individuals living in regional, rural, and remote Australia is proposed.

A preliminary search of PubMed, Joanna Briggs Institute (JBI) Evidence Synthesis, the Cochrane Database of Systematic Reviews, and PROSPERO was conducted. No reviews which specifically examined the barriers and facilitators in relation to accessing and utilising mental health services across Australia were identified. One review with a global scope specifically focussed on attitudinal barriers and identified four themes—stoicism, stigma, distrust, and meaning—which impacted help-seeking in individuals who reside in rural Australia, Canada, and the United States of America [30]. Other reviews on related topics have exclusively focussed on the mental health issues and/or burden of Australian farming communities [31], as well as adolescent [32], at-risk [33], male [34], and Indigenous populations [35, 36]. Some reviews have explored the delivery of services and workforce issues [37], or trends in Australian rural mental health research [38]. Kennedy et al. [39], using government-collected suicide and coronial data, found that, compared to non-farming-related suicides in Australia, farming-related suicides were less likely to be associated with receiving a mental health diagnosis and seeking GP delivered mental health treatment in the six weeks prior to death. Coupled with the recommendations of the Royal Commission, these data put forward the urgency in understanding the issues associated with accessing mental health services for regional, rural, and remote Australians.

Owing to the cross-discipline diversity of emerging literature pertaining to this research enquiry, we propose to undertake a scoping review. A scoping review aims to identify and map the literature, recognise key concepts, identify gaps in the evidence, and ascertain the nature and source of the literature, which may help to inform clinical practice, healthcare planning, and policy [40]. The aims of the current scoping review are to: (1) explore the barriers and facilitators to accessing and utilising mental health services for Australians living in regional, rural, and remote areas; and (2) determine the relationship between the barriers and facilitators experienced in accessing and utilising mental healthcare by geographical location.

## Methods and analysis

This scoping review protocol is concordant with the framework put forward by Arksey and O'Malley [41], as well as the developments to the original framework put forward by other researchers [40, 42, 43]. The following processes will be applied and are described in detail below:

- Stage 1: Identifying the research question

- Stage 2: Identifying relevant studies

- Stage 3: Study selection

- Stage 4: Charting the data

- Stage 5: Collating, summarising, and reporting the results.

This scoping review protocol complies, where applicable, with the Preferred Reporting Items for Systematic Reviews and Meta-Analysis Protocols (PRISMA-P) checklist [44] (see S1 Checklist).

### Stage 1: Identifying the research question

The authors undertook an interactive process of refining the research question based on the available literature. At this point, we also considered the usefulness in recording the geographical location of the barriers and facilitators identified. Accordingly, at this first stage, the following research questions were developed:

1. What is known about the barriers and facilitators experienced by regional, rural, and remote Australians in accessing and utilising mental health services?

2. What is the relationship between the barriers and facilitators in accessing and utilising mental healthcare and geographical location?

## Definitions

In Australia, the Australian Statistical Geography Standard Remoteness Area (ASGS-RA) is a nationally consistent approach used to objectively describe geographical access. The ASGS-RA denotes five areas of remoteness: Major Cities (RA1), Inner Regional (RA2), Outer Regional (RA3), Remote (RA4), and Very Remote Australia (RA5) [45]. From 2022, the Australian Government Department of Health is moving to the Modified Monash Model (MMM). The MMM is based on the ASGS-RA and accounts for both remoteness and population size, and includes the following seven classifications: MM1 Metropolitan Areas, MM2 Regional Centres, MM3 Large Rural Towns, MM4 Medium Rural Towns, MM5 Small Rural Towns, MM6 Remote Communities, and MM7 Very Remote Communities [13]. Under this classification, areas classified as MM2-MM7 will be classified as rural.

For the purposes of this review and akin to Stroud and Lockwood [34], barriers refer to obstacles that obstruct the utilisation of mental health services provision—this may also include factors which prevent the quality and level of warranted services being accessed [46]. Barriers may include but are not limited to confidentiality concerns, fear of stigma, and poor mental health literacy. Facilitators are the factors that allow the uptake of mental health services to occur, this may also include factors that permit the appropriate amount/quality of care to be received [46]. Facilitators may include but are not limited to telehealth availability, free/low-cost services, and appointment timeliness. Access refers to the uptake of mental health services and is denoted as belonging to aspatial (i.e., affordability, acceptability, and accommodation) or spatial (i.e., availability and accessibility) dimensions [6]. Utilisation infers access, but additionally involves the development of a healthcare plan during and healthcare encounter and its consequential implementation [11].

## Stage 2: Identifying relevant studies

The population/concept/context (PCC) framework was used to develop the inclusion criteria for this scoping review (see Table 1).

## Inclusion criteria

Eligible articles include those which describe barriers and/or facilitators to accessing and/or utilising mental health services in regional, rural, and remote Australia for patients/individuals with mental health issues/concerns and healthcare professionals providing mental health services. We are interested in examining literature that reports individuals over the lifespan (i.e., children, adolescents, and adults) and as such, no age restrictions will be imposed.

Only studies that have been conducted in Australia within the last decade (2012-present) are eligible for review. This is because significant changes have occurred within the Australian mental health system during this time in line with the Roadmap for National Mental Health Reform (the Roadmap) [47]. The Roadmap was designed to improve access to high quality services and supports, as well as enhance mental health and wellbeing, focus on prevention, early detection, and intervention, emphasise person-centred approaches, and progress social and economic participation of individuals with mental illness [47].

This review will include quantitative studies that are experimental or quasi-experimental, observational, case-control, and cross-sectional study designs. Qualitative research, as well as mixed-methods, case-series, and case-reports will also be included. Non-peer reviewed articles, including grey literature (e.g., government reports, policy statements, issues papers, dissertations, and fact sheets) will be considered for inclusion (see S1 File for grey literature sources). Only evidence which has been published in the English language will be eligible.

*Search strategy and databases*. A comprehensive list of primary search terms and their variants were derived from the research question and using the PCC framework. The indicative search strategy was developed with the support of a health librarian (see Table 2) and the final search strategy will be evaluated by the Peer Review of Electronic Search Strategies (PRESS) checklist [48]. The final search strategy will be translated (using applicable index terms, Boolean operators, truncation and wildcard symbols) and implemented in health, medical, and psychology databases including Medline Complete, EMBASE, PsycINFO, Scopus, and Cumulative Index of Nursing and Allied Health Literature (CINAHL). Websites of government departments, primary health networks, peak bodies/agencies, key grey literature databases will be searched to capture grey literature (see S1 File). A date of 2012-current will be applied to filter the search yield. 'Snowball' searching of additional sources of evidence will be undertaken via searching the reference lists of identified studies, citation tracking, and/or existing networks.

**Table 1. PCC (population/concept/context).**

|   | Inclusion | Example | Exclusion |
|---|---|---|---|
| **P** | Patients/individuals with mental health issues/concerns of any age | Diagnosed mental disorders:<br>• Schizophrenia (spectrum) and other psychotic disorders<br>• Depressive disorders<br>• Bipolar and related disorders<br>• Anxiety disorders<br>• Obsessive-compulsive and related disorders<br>• Trauma- and stressor-related disorders<br>• Somatic and related disorders<br>• Dissociative disorders<br>• Feeding and eating disorders<br>• Disruptive, impulse control, and conduct disorders<br>• Substance-related and additive disorders<br>• Personality disorders<br>Mental health issues:<br>• Psychological distress indicated via validated measure<br>• "At-risk" groups (e.g., where mental health service provision has been sought/warranted but a diagnosis has not yet been made)<br>• Mental disorders not otherwise specified<br>Part of mental health/community service (current or past client):<br>• Adult mental health service<br>• Child and adolescent mental health service<br>• Community mental health organisation<br>• Be a carer of an individual who has a diagnosed mental disorder/mental health issue/part of a mental health or community service | Neurodevelopmental disorders<br>Elimination disorders<br>Sleep-wake disorders<br>Sexual dysfunctions<br>Gender dysphoria<br>Neurocognitive disorders<br>Paraphilic disorders |
|   | Healthcare providers providing diagnostic/assessment/treatment for mental health issues | • Medical specialists (e.g., general practitioners and psychiatrists)<br>• Allied health professionals (e.g., psychologists, social workers, counsellors)<br>• Nurses and nurse practitioners<br>• Drug and alcohol workers<br>• Community mental health workers (i.e., workers who provide social/housing/occupational support)<br>• Peer-workers<br>• Personal helpers and mentors<br>• Pharmacists | Healthcare providers who do not specifically diagnose/assess/treat individuals with mental health issues |

(*Continued*)

**Table 1.** (Continued)

| | Inclusion | Example | Exclusion |
|---|---|---|---|
| C | Barriers | Obstacles that obstruct the uptake of mental health services or factors that prevent the quality/level of care being accessed<br>• Confidentiality concerns<br>• Fear of stigma<br>• Poor mental health literacy<br>• Geographic isolation<br>• Limited appointment availability<br>• High cost of service | Factors that are not considered to be barriers |
| | Facilitators | Factors that permit the uptake of mental health services or factors that allow the appropriate amount/quality of care to be received:<br>• Telehealth availability<br>• Free/low cost of service<br>• Appointment timeliness<br>• Safe and supportive environment<br>• Culturally competent health providers<br>• Mentors to assist with system navigation | Factors that are not considered to be facilitators |
| | Access factors | Factors that measure of the alignment between healthcare provider/services and the characteristics/expectations of clients:<br>• Aspatial dimensions (i.e., affordability, acceptability, and accommodation)<br>• Spatial dimensions (i.e., availability and accessibility) | Factors that are not considered to be related to access |
| | Utilisation factors | Factors that affect the utilisation of healthcare services, including the implementation of subsequent healthcare encounters:<br>• Effective information exchange<br>• Satisfactory negotiation of a healthcare plan<br>• Interpersonal relationship between the healthcare provider and the patient | Factors that are not considered to be related to the utilisation of mental health services |
| | Mental health services | • Services provided by hospitals (public and private)<br>• Community-based services (i.e., Acute Community Intervention Service [ACIS], community care units (CCUs), Prevention and Recovery Centres (PARCs), and outpatient clinical treatment<br>• Mental Health Community Support Services (MHCSS) (e.g., services that are operated by non-government organisations)<br>• Specialist mental health services (e.g., services provided specifically for individuals with certain mental health needs)<br>• Outreach services<br>• Early intervention services embedded within schools | All other health services, mental health programs, health promotion initiatives |
| C | Regional, rural, and remote areas of Australia | Areas classified as regional, rural, or remote Australia according to the MMM:<br>• MM2 Regional Centres<br>• MM3 Large Rural Towns<br>• MM4 Medium Rural Towns<br>• MM5 Small Rural Towns<br>• MM6 Remote Communities<br>• MM7 Very Remote Communities | Areas classified as a Major City in Australia according to the MMM:<br>• MM1 Metropolitan Areas |

## Stage 3: Study selection

One reviewer will apply the search strategy, consolidate the results, and remove duplicates. Endnote X9 [49] and Covidence [50] will be used for reference management. Screening of the titles and/or abstracts, as well as full-text articles will be independently conducted by two reviewers. For grey literature, the titles of articles/information and any associated short text will be searched in the first instance, and then if applicable, the full text will be assessed for eligibility. Reasons for exclusion will be provided for the full-text screening stage. If there is discrepancy between the two reviewers concerning eligibility, the issue will be discussed between the two reviewers and, if necessary, a third reviewer will be consulted to make a final decision regarding consensus.

**Table 2. Indicative search strategy for medline complete via EBSCO.**

| Search line | Seach terms |
|---|---|
| #1 | (MH "mental disorder"+) |
| #2 | (TI "mental health") |
| #3 | (TI "mental illness*") |
| #4 | (TI "mental disorder*") |
| #5 | (TI "mental distress") |
| #6 | (TI "psychiatric illness*") |
| #7 | (TI depression) |
| #8 | (TI anxiety) |
| #9 | (TI psychosis) |
| #10 | (TI "substance use") |
| #11 | (TI "substance abuse") |
| #12 | (TI "drug use") |
| #13 | (TI "drug abuse") |
| #14 | (TI "drug addiction") |
| #15 | (TI "personality disorder*") |
| #16 | (TI "eating disorder*") |
| #17 | (TI schizophrenia) |
| #18 | (TI suicid*) |
| #19 | (#2 OR #3 #4 OR #5 OR #6 OR #7 OR #8 OR #9 OR #10 OR #11 OR #12 OR #13 OR #14 OR #15 OR #16 OR #17 OR #18) |
| #20 | (AB "mental health") |
| #21 | (AB "mental illness*") |
| #22 | (AB "mental disorder*") |
| #23 | (AB "mental distress") |
| #24 | (AB "psychiatric illness*") |
| #25 | (AB depression) |
| #26 | (AB anxiety) |
| #27 | (AB psychosis) |
| #28 | (AB "substance use") |
| #29 | (AB "substance abuse") |
| #30 | (AB "drug use") |
| #31 | (AB "drug abuse") |
| #32 | (AB "drug addiction") |
| #33 | (AB "personality disorder*") |
| #34 | (AB "eating disorder*") |
| #35 | (AB schizophrenia) |
| #36 | (AB suicid*) |
| #37 | (#20 OR #21 OR #22 OR #23 OR 24 #25 OR #26 OR #27 OR #28 OR #29 OR #30 OR #31 OR #32 OR #33 OR #34 OR #35 OR #36) |
| #38 | (#1 OR #19 OR #37) |
| #39 | (TI barrier*) |
| #40 | (TI obstacle*) |
| #41 | (TI challeng*) |
| #42 | (TI facilitat*) |
| #43 | (TI enabl*) |
| #44 | (TI "help seek*") |
| #45 | (TI "help-seek*") |

(*Continued*)

**Table 2.** (Continued)

| Search line | Seach terms |
|---|---|
| #46 | (#39 OR #40 OR #41 OR #42 OR #43 OR #44 OR #45) |
| #47 | (AB barrier*) |
| #48 | (AB obstacl*) |
| #49 | (AB challeng*) |
| #50 | (AB facilitat*) |
| #51 | (AB enabl*) |
| #52 | (AB "help seek*") |
| #53 | (AB "help-seek*") |
| #54 | (#47 OR #48 OR #49 OR #50 OR #51 OR #52 OR #53) |
| #55 | (#46 OR #54) |
| #56 | (MH "rural health services"+) |
| #57 | (TI regional) |
| #58 | (TI rural) |
| #59 | (TI remote) |
| #60 | (#57 OR #58 OR #59) |
| #61 | (AB regional) |
| #62 | (AB rural) |
| #63 | (AB remote) |
| #64 | (#61 OR #62 OR #63) |
| #65 | (#56 OR #60 OR #64) |
| #66 | (MH Australia+) |
| #67 | (TI Australia) |
| #68 | (TI Victoria) |
| #69 | (TI "New South Wales") |
| #70 | (TI NSW) |
| #71 | (TI Queensland) |
| #72 | (TI "Northern Territory") |
| #73 | (TI NT) |
| #74 | (TI "South Australia") |
| #75 | (TI "Western Australia) |
| #76 | (TI "Australian Capital Territory") |
| #77 | (TI Tasmania) |
| #78 | (#67 OR #68 OR #69 OR #70 OR #71 OR #72 OR #73 OR #74 OR #75 OR #76 OR #77) |
| #79 | (AB Australia) |
| #80 | (AB Victoria) |
| #81 | (AB "New South Wales") |
| #82 | (AB NSW) |
| #83 | (AB Queensland) |
| #84 | (AB "Northern Territory") |
| #85 | (AB NT) |
| #86 | (AB "South Australia") |
| #87 | (AB "Western Australia) |
| #88 | (AB "Australian Capital Territory") |
| #89 | (AB Tasmania) |
| #90 | (#79 OR #80 OR #81 OR #82 OR #83 OR #84 OR #85 OR #86 OR #87 OR #88 OR #89) |
| #91 | (#66 OR #78 OR #90) |
| #92 | (#38 AND #55 AND #65 AND #91) |

## Stage 4: Charting the data

Relevant data will be extracted from eligible studies and imputed into a charting form developed for this review. The charting form will include information on study descriptors (i.e., author/year, study objective, study design, and location), population (i.e., sample size, characteristics, mental health issue/condition and assessment method, health care provider), concept (i.e., barriers, facilitators, access factors, utilisation factors, mental health care), context (i.e., regional/rural/remote areas of Australia), and results (i.e., key results and summary of findings). In addition, and similar to a review by Beks et al. [51, 52], the geographical location of where the barriers and/or facilitators are identified will be analysed using a Geographical Information System (GIS) program, ArcGIS ArcMap 10.6.1 (ESRI, CA, USA). The GIS program permits the visualisation, of geographical data on a map. In order to conduct the geographical analysis, the location of the study (i.e., service or client/service provider group of interest) will be charted. If these data are not available, the corresponding author will be contacted to ascertain study location. In cases where study location is unavailable, the location corresponding to the first author's affiliation will be used. Using geographical coordinates, the appropriate MMM category will be assigned to each of the charted locations based on the ArcGIS ArcMap 10.6.1 (ESRI, CA, USA). The location of studies will be stratified using the MMM. Adopting the MMM has explicit and contemporary relevance to policy, including through the workforce (e.g., Department of Health programs are changing to the MMM), research translation (e.g., 2020 Rapid Applied Research Translation Grant Opportunity), and service delivery (e.g., Medicare rebates on psychology telehealth consultations) [16]. Further, its application is important for the consistency of reviews in health research going forward [16]. These data will then be exported to STATA and summary statistics will be produced to analyse the relationships between the geographical data and the identified barriers and facilitators. The use of the MMM will allow insights into barriers and facilitators pertinent to remoteness and population size—allowing comparisons across and within MMM categories to be made, regardless of the heterogeneity of the population itself.

The charting form will be independently tested by two reviewers to ensure that it is suitable to answer the research question. This piloting will require the two reviewers to independently chart data from five studies and consequently hold a consensus meeting with the supervising author to deliberate results and resolve any discrepancies. Any modifications for the charting form will be detailed within the scoping review. One reviewer will then chart the data using the predetermined charting form (see Table 3) and the second reviewer will validate the extracted information. Any discrepancies will be discussed, and a third reviewer will provide a consensus if needed.

## Stage 5: Collating, summarising, and reporting the results

A flow diagram will document the flow of the search yield and eligibility process. Characteristics of the included studies will be discussed. Subsequent to charting the data, key findings will

**Table 3. Indicative charting form for data extraction.**

| Descriptors | | | | Population | | | | Concept | | | | | Context | Results |
|---|---|---|---|---|---|---|---|---|---|---|---|---|---|---|
| Author & year | Study objective | Study design | Location | Sample size | Charact-eristics (e.g., age, sex) | Mental health condition/ issue and assessment method | Health-service provider (e.g., type/ role) | Barriers | Facilitators | Access factors (i.e., aspatial or spatial dimensions) | Utilisation factors | Mental health care service) | Regional/ rural/ remote areas of Australia | Summary of findings |

be summarised according to the research question using a narrative account. A discussion of the relationships identified via the geographical analysis will also be included. Findings will be presented in text and may also be displayed using tables and maps, as appropriate. The narrative account will include a descriptive numerical summary of the nature and distribution of the included studies (e.g., study design, year of publication, characteristics of the study populations, and geographical area) and presentation of key themes as identified in the charting process. This may include presenting information according to similar barriers/facilitators in relation to access and utilisation factors; remoteness categories; findings specific to states/territories; or other relevant grouping/s.

## Discussion and dissemination

This scoping review will provide a contemporary account exploring the barriers and facilitators to accessing and utilising mental health services within the context of regional, rural, and remote Australia. The spatial distribution will provide a visualisation of the study location and corresponding barriers and facilitators as documented by the eligible research—affording insights into such factors that are pertinent to remoteness and population size. This scoping review may be limited by the available literature, which may not reflect the barriers and facilitators of specific groups or areas within the wider population. In addition, the use of the MMM may preclude a nuanced analysis of the geographical context. Nonetheless, this scoping review may aid in improving mental health outcomes by documenting the factors that may contribute to inequalities in service access and utilisation, and mapping where the research is being conducted. Together, this information may help to provide knowledge on gaps in research activity and service delivery, including resource allocation and workforce shortages, as well as patterns of mental health service utilisation. Alston et al. [53] identified that a lack of evidence in the rural context impedes policymakers from making evidence-informed decisions to enhance the health of rural populations. As such, it is anticipated that the results of this scoping review will be relevant to policymakers at a national, state, and local level and may also be utilised by healthcare providers. Results will be disseminated via publication in a scientific peer-reviewed journal and presented at a conference.

## Supporting information

**S1 Checklist. PRISMA-P 2015 checklist.**
(DOCX)

**S1 File. Grey literature information sources.**
(DOCX)

## Acknowledgments

The authors would like to acknowledge Jill Stephens and Blair Kelly from Deakin University for their assistance in developing the search strategy.

## Author Contributions

**Conceptualization:** Bianca E. Kavanagh, Hannah Beks, Vincent L. Versace, Shae E. Quirk, Lana J. Williams.

**Methodology:** Bianca E. Kavanagh, Hannah Beks, Vincent L. Versace, Shae E. Quirk, Lana J. Williams.

**Project administration:** Bianca E. Kavanagh.

**Writing – original draft:** Bianca E. Kavanagh.

**Writing – review & editing:** Bianca E. Kavanagh, Hannah Beks, Vincent L. Versace, Shae E.
  Quirk, Lana J. Williams.

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
