## [Decision Letter · Decision Letter 0]

19 Jul 2022

PONE-D-22-05190

Exploring the barriers and facilitators to accessing and utilising mental health care in regional, rural, and remote Australia: A scoping review protocol

PLOS ONE

Dear Dr. Kavanagh,

Thank you for submitting your manuscript to PLOS ONE. After careful consideration, we feel that it has merit but does not fully meet PLOS ONE’s publication criteria as it currently stands. Therefore, we invite you to submit a revised version of the manuscript that addresses the points raised during the review process.

I would like to sincerely apologise for the delay you have incurred with your submission. It has been exceptionally difficult to secure reviewers to evaluate your study. We have now received two completed reviews; the comments are available below. The reviewers have raised significant scientific concerns about the study that need to be addressed in a revision.

Please revise the manuscript to address all the reviewer's comments in a point-by-point response in order to ensure it is meeting the journal's publication criteria. Please note that the revised manuscript will need to undergo further review, we thus cannot at this point anticipate the outcome of the evaluation process.

We look forward to receiving your revised manuscript.

Kind regards,

Miquel Vall-llosera Camps, Ph.D.

Staff Editor

PLOS ONE

“The authors declare that they have no competing interests.”

Reviewers' comments:

Reviewer's Responses to Questions

**Comments to the Author**

1. Does the manuscript provide a valid rationale for the proposed study, with clearly identified and justified research questions?

Reviewer #1: Partly

Reviewer #2: Yes

2. Is the protocol technically sound and planned in a manner that will lead to a meaningful outcome and allow testing the stated hypotheses?

Reviewer #1: Partly

Reviewer #2: Yes

3. Is the methodology feasible and described in sufficient detail to allow the work to be replicable?

Reviewer #1: No

Reviewer #2: Yes

4. Have the authors described where all data underlying the findings will be made available when the study is complete?

Reviewer #1: Yes

Reviewer #2: No

5. Is the manuscript presented in an intelligible fashion and written in standard English?

Reviewer #1: Yes

Reviewer #2: Yes

6. Review Comments to the Author

You may also provide optional suggestions and comments to authors that they might find helpful in planning their study.

Reviewer #1: 1. My major concern relates to the second research question and the use of GIS mapping in the review. To my mind, the authors have not provided a sound rationale nor sufficient detail for the use of GIS mapping in their scoping review. For example, how will this facilitate exploration of the relationship between barriers/facilitators and geographic location as outlined in the second research question. How this second research question will be answered is not sufficiently addressed in the discussion either, which claims the paper will provide a synthesis. Research question 2, however, suggests that the paper seeks to provide more than a synthesis, but also an analysis.

It is also not clear to me whether GIS in and of itself allows the identification of relationships between variables or if it merely lists variables (line 292). For readers not familiar with this method, the authors need to provide further explanation and to explain how analysis will be conducted given the possible relationships between individual, institutional, socio-cultural, and political factors in determining barriers or facilitators to access.

Also, if mapping is stratified according to the classification system chosen by the authors (MMM), how will this permit an understanding of the heterogeneity of place in relation to regional, rural and remote communities. As the authors point out, there is considerable diversity across Australia in terms of population density, service distribution, environmental factors, employment and educational opportunities, social capital and resources. Even within areas classified as remote, for example, there is considerable variation with regards to the prevalence of mental health and suicide. Does mapping by remoteness category risk erasing these differences (i.e. viewing all small rural towns as the same). Moreover, will mapping allow for the capturing of issues such as drought, flood or other environmental issues, economic downturns, policy changes etc. that may have impacted mental health, mental health access, or the provision of services within specific communities? Again, further clarification of this method of analysis in relation to the research question and a discussion of potential limitations of this method is needed.

2. Following on from the above, the authors state that this information will be useful for policymakers and healthcare providers. However, findings are dependent on the literature that will be extracted in this review (which is dependent on the research that has been carried out in Australia in the last decade and which is subject to the interests of individual researchers, funding bodies etc? For example, there may be a wealth of studies on farming communities, on Aboriginal and Torres Strait Islander communities, or on young people while other studies may simply focus on rates of mental health literacy between rural and urban communities). Again, this seems to me like a potential limitation that needs to be acknowledged, especially given the relatively small body of rural mental health research conducted in Australia and the focus of much of it in recent years on individual-level factors and specific at-risk populations rather than on ecological or place-based studies.

3. The authors need to explain how they will deal with the use of different remoteness classification systems likely to have been used within the research literature.

4. More information is needed on how the narrative synthesis will be conducted. This term is generic and provides little clarity on actual methods. The authors need to be more transparent and explicit about the method of analysis they will use, especially in the likelihood that diverse data will be used to construct a narrative about barriers and/or facilitators.

5. The discussion section is very short on information. As well as a discussion of challenges/limitations, the authors might also consider discussing how this information may be potentially useful to policymakers, particularly considering the broad range of issues highlighted in the introduction. Some discussion of the potential benefits would be good here.

6. I was unable to open document S2 reporting on the search strategy for the grey literature and so have been unable review this.

Reviewer #2:

Dear authors,

Thank you for the opportunity to review this exciting scoping review protocol. As a protocol, this manuscript is highly relevant to the Australian contexts, and I would like to see to the results might also be applicate to other settings. As a protocol, there is not much new insights can be provided, however, when the review is completed this scoping review will offer meaningful contribution to the mental health service delivery for rural and remote communities in Australia, and perhaps to other settings as well. I have some minor comments to think about regarding the current protocol:

Introduction:

• While the disparities across remoteness are covered in the introduction, it is important to state clearly some data around hospital presentation, availability of medicines, availability of psychologist/clinical psych, and mental health nurses.

Methods:

• Review questions – two questions being asked in this scoping review may require two different search strategies and/or different analysis approaches. Please clarify!

7. PLOS authors have the option to publish the peer review history of their article (what does this mean?). If published, this will include your full peer review and any attached files.

Reviewer #1: No

Reviewer #2: No

---

## [Author Response · Author response to Decision Letter 0]

25 Aug 2022

Professor Emily Chenette

PLOS ONE

Editor in Chief

RE: Revision of manuscript id PONE-D-22-05190 “Exploring the barriers and facilitators to accessing and utilising mental health services in regional, rural, and remote Australia: A scoping review protocol”.

We thank the reviewers for their feedback and for the opportunity to revise our manuscript. We have addressed all queries raised by the reviewers. In this document, the reviewer’s queries are presented in boldface and our responses are in regular text, with changes to the manuscript text indicated in italics and quotation marks. In the revised manuscript submission, changes from the original manuscript are displayed using tracked changes.

Review comments to authors:

Reviewer: 1

1. My major concern relates to the second research question and the use of GIS mapping in the review. To my mind, the authors have not provided a sound rationale nor sufficient detail for the use of GIS mapping in their scoping review. For example, how will this facilitate exploration of the relationship between barriers/facilitators and geographic location as outlined in the second research question. How this second research question will be answered is not sufficiently addressed in the discussion either, which claims the paper will provide a synthesis. Research question 2, however, suggests that the paper seeks to provide more than a synthesis, but also an analysis.

We have now provided a rationale in the introduction for the spatial analysis/use of GIS mapping. We have also provided more detail in the method and discussion sections pertaining to the information that will be charted and how this will answer the research questions. Research question 2 does involve an analysis of the spatial data corresponding with the location of the eligible studies—further detail has been provided in the method section. 

Amendments to text as follows:

“Research that documents inequalities in accessing and utilising mental health services may moderate the disease burden experienced by regional, rural, and remote populations and assist in equitable resource allocation (24). Spatial distribution may be a useful tool in assessing where the barriers and facilitators to accessing and utilising mental health services exist across Australia, particularly considering that the distance and spatial availability of mental health services reduce their utilisation (25-27). Spatial distribution permits visualisation and may help to identify priority areas of where barriers need to be overcome and provide examples of facilitators that may be implemented in other areas. Spatial distribution may therefore aid in policy making, especially in relation to resource allocation, workforce shortages, and postulation of service utilisation patterns (27, 28).” (Lines 139-148)

“In order to conduct the geographical analysis, the location of the study (i.e., service or client/service provider group of interest) will be charted. If these data are not available, the corresponding author will be contacted to ascertain study location. In cases where study location is unavailable, the location corresponding to the first author’s affiliation will be used. Using geographical coordinates, the appropriate MMM category will be assigned to each of the charted locations based on the ArcGIS ArcMap 10.6.1 (ESRI, CA, USA).” (Lines 314-319)

“These data will then be exported to STATA and summary statistics will be produced to analyse the relationships between the geographical data and the identified barriers and facilitators. The use of the MMM will allow insights into barriers and facilitators pertinent to remoteness and population size—allowing comparisons across and within MMM categories to be made, regardless of the heterogeneity of the population itself.” (Lines 325-329)

“The spatial distribution will provide a visualisation of the study location and corresponding barriers and facilitators as documented by the eligible research—affording insights into such factors that are pertinent to remoteness and population size.” (Lines 359-361)

It is also not clear to me whether GIS in and of itself allows the identification of relationships between variables or if it merely lists variables (line 292). For readers not familiar with this method, the authors need to provide further explanation and to explain how analysis will be conducted given the possible relationships between individual, institutional, socio-cultural, and political factors in determining barriers or facilitators to access.

Although the GIS does allow the identification of relationships between variables, we will be using use the GIS for mapping the study location only. These data will then be imputed into STATA to explore patterns between barriers/facilitators and geographic context. Please refer to the above response regarding the details of the analysis and relationships between the identified barriers and facilitators. The individual, institutional, sociocultural, and political factors will be discussed within the narrative account of the identified barriers and facilitators. 

Also, if mapping is stratified according to the classification system chosen by the authors (MMM), how will this permit an understanding of the heterogeneity of place in relation to regional, rural and remote communities. As the authors point out, there is considerable diversity across Australia in terms of population density, service distribution, environmental factors, employment and educational opportunities, social capital and resources. Even within areas classified as remote, for example, there is considerable variation with regards to the prevalence of mental health and suicide. Does mapping by remoteness category risk erasing these differences (i.e. viewing all small rural towns as the same). Moreover, will mapping allow for the capturing of issues such as drought, flood or other environmental issues, economic downturns, policy changes etc. that may have impacted mental health, mental health access, or the provision of services within specific communities? Again, further clarification of this method of analysis in relation to the research question and a discussion of potential limitations of this method is needed.

We agree that there is considerable heterogeneity across Australia. However, there are also similarities between areas. For instance, Versace et al (2021) found consistently low socioeconomic status across MM2-MM7 categories. We believe that a nuanced analysis that accounts for specific issues (e.g., environmental issues, economic changes, service provision specific to communities) is beyond the scope of the current study, and that in the first instance, providing a narrative account and spatial analysis of the current topic is warranted. Specific issues that may lead to barriers/facilitators may be discussed within the context of the narrative account. The limitations of the current method have now been described in the discussion section. 

Amendments to text as follows:

“This scoping review may be limited by the available literature, which may not reflect the barriers and facilitators of specific groups or areas within the wider population. In addition, the use of the MMM may preclude a nuanced analysis of the geographical context.” (Lines 361-364)

2. Following on from the above, the authors state that this information will be useful for policymakers and healthcare providers. However, findings are dependent on the literature that will be extracted in this review (which is dependent on the research that has been carried out in Australia in the last decade and which is subject to the interests of individual researchers, funding bodies etc? For example, there may be a wealth of studies on farming communities, on Aboriginal and Torres Strait Islander communities, or on young people while other studies may simply focus on rates of mental health literacy between rural and urban communities). Again, this seems to me like a potential limitation that needs to be acknowledged, especially given the relatively small body of rural mental health research conducted in Australia and the focus of much of it in recent years on individual-level factors and specific at-risk populations rather than on ecological or place-based studies.

We agree that this scoping review is limited by the available evidence—as is the case with all reviews. We have now acknowledged this as a limitation (see the above amendment). We also believe that the use of grey literature will allow us to include information that is less restricted by the interests of individual researchers and funding bodies, as we expect to identify peak body documents and submissions to parliament about specific rural mental health issues. 

3. The authors need to explain how they will deal with the use of different remoteness classification systems likely to have been used within the research literature.

We will be charting the study’s location (or the location of the first author, if applicable) and assigning the MMM classification system based on geographical coordinates. Using this method, we will only be dealing with one classification system. 

4. More information is needed on how the narrative synthesis will be conducted. This term is generic and provides little clarity on actual methods. The authors need to be more transparent and explicit about the method of analysis they will use, especially in the likelihood that diverse data will be used to construct a narrative about barriers and/or facilitators.

In line with Arksey and O’Malley’s guidelines on collating, summarising, and reporting the results, we have changed the wording from ‘synthesis’ to ‘narrative account’. More detail about what the narrative account will include has now been provided. However, the presentation of results (i.e., inclusion of specific information) will depend on the information that is obtained during the charting process. 

Amendments to text as follows:

“The narrative account will include a descriptive numerical summary of the nature and distribution of the included studies (e.g., study design, year of publication, characteristics of the study populations, and geographical area) and presentation of key themes identified in the charting process. This may include presenting information according to similar barriers/facilitators in relation to access and utilisation factors; remoteness categories; findings specific to states/territories; or other relevant grouping/s.” (Lines 348-353)

5. The discussion section is very short on information. As well as a discussion of challenges/limitations, the authors might also consider discussing how this information may be potentially useful to policymakers, particularly considering the broad range of issues highlighted in the introduction. Some discussion of the potential benefits would be good here.

In addition to the limitations of the review now being provided (see reviewer 1 comment 1 amendments to text), information on the benefits of this review and how it may aid in policymaking has now been included.

Amendments to text as follows:

“Nonetheless, this scoping review may aid in improving mental health outcomes by documenting the factors that may contribute to inequalities in service access and utilisation, and mapping where the research is being conducted. Together, this information may help to provide knowledge on gaps in research activity and service delivery, including resource allocation and workforce shortages, as well as patterns of mental health service utilisation. Alston et al. (55) identified that a lack of evidence in the rural context impedes policymakers from making evidence-informed decisions to enhance the health of rural populations.” (Lines 364-371)

6. I was unable to open document S2 reporting on the search strategy for the grey literature and so have been unable review this.

We apologise that you were unable to open the S2 document. We have re-uploaded this document with this submission. 

Reviewer #2:

Dear authors,

Thank you for the opportunity to review this exciting scoping review protocol. As a protocol, this manuscript is highly relevant to the Australian contexts, and I would like to see to the results might also be applicate to other settings. As a protocol, there is not much new insights can be provided, however, when the review is completed this scoping review will offer meaningful contribution to the mental health service delivery for rural and remote communities in Australia, and perhaps to other settings as well. I have some minor comments to think about regarding the current protocol:

Introduction:

1. While the disparities across remoteness are covered in the introduction, it is important

to state clearly some data around hospital presentation, availability of medicines, availability of psychologist/clinical psych, and mental health nurses.

We have amended the previous information on the rural health workforce to include specific data on the number of psychiatrists, mental health nurses and psychologists. We have also added information on the rate of mental health-related emergency presentations. However, as we have decided to refine our research questions to mental health services, rather than the broader concept of mental health ‘care’ (see note below) we have not provided information on the availability of medicines. 

Amendments to text as follows:

“This is demonstrated by a two-fold increased rate of mental health-related emergency presentations (4)…” (Lines 81-82)

“In 2020, there was a greater number of clinical full-time equivalent mental health specialists per/100,000 population working in Major Cities compared to other remoteness areas (i.e., 14.8 vs 2.6-6.5 psychiatrists, 97.0 vs 31.5-83.8 mental health nurses, 95.9 vs 23.5-59.9 psychologists).” (Lines 110-113)

Methods:

2. Review questions – two questions being asked in this scoping review may require two different search strategies and/or different analysis approaches. Please clarify!

The first research question pertains to understanding what the barriers and facilitators are, while the second research question seeks to map the research in order to analyse the relationship between the barriers and facilitators and geographical location. Both research questions can be answered with the same search strategy, and will be discussed in a narrative account. The geographic locations (research question 2) will additionally be presented in a map. 

Other author changes:

In addition to the above changes, we have refined our scoping review to be focussed on mental health ‘services’ rather than ‘care’. This was because the concept of ‘care’ involves preventative and promotion initiatives, of which include the general population, rather than being specific to those with mental health issues. We have also removed the ‘key results’ column from the charting form, as this repeated information that was included in the ‘concept’ and ‘summary of findings’ column. These changes have been reflected throughout the manuscript.

---

## [Decision Letter · Decision Letter 1]

21 Nov 2022

Exploring the barriers and facilitators to accessing and utilising mental health services in regional, rural, and remote Australia: A scoping review protocol

PONE-D-22-05190R1

Dear Dr. Kavanagh,

We’re pleased to inform you that your manuscript has been judged scientifically suitable for publication and will be formally accepted for publication once it meets all outstanding technical requirements.

Kind regards,

Ejaz Ahmad Khan, M.D, MPH, FFPH

Academic Editor

PLOS ONE

Additional Editor Comments (optional):

Reviewers' comments:

Reviewer's Responses to Questions

**Comments to the Author**

1. Does the manuscript provide a valid rationale for the proposed study, with clearly identified and justified research questions?

Reviewer #1: Yes

2. Is the protocol technically sound and planned in a manner that will lead to a meaningful outcome and allow testing the stated hypotheses?

Reviewer #1: Yes

3. Is the methodology feasible and described in sufficient detail to allow the work to be replicable?

Reviewer #1: Yes

4. Have the authors described where all data underlying the findings will be made available when the study is complete?

Reviewer #1: Yes

5. Is the manuscript presented in an intelligible fashion and written in standard English?

Reviewer #1: Yes

6. Review Comments to the Author

You may also provide optional suggestions and comments to authors that they might find helpful in planning their study.

Reviewer #1: I am satisfied that these revisions address the issues outlined by peer reviewers and recommend the manuscript be accepted for publication.

7. PLOS authors have the option to publish the peer review history of their article (what does this mean?). If published, this will include your full peer review and any attached files.

Reviewer #1: No

---

## [Editor Report · Acceptance letter]

1 Dec 2022

PONE-D-22-05190R1 

Exploring the barriers and facilitators to accessing and utilising mental health services in regional, rural, and remote Australia: A scoping review protocol 

Dear Dr. Kavanagh:

I'm pleased to inform you that your manuscript has been deemed suitable for publication in PLOS ONE. Congratulations! Your manuscript is now with our production department. 

Kind regards, 

on behalf of

Dr. Ejaz Ahmad Khan 

Academic Editor

PLOS ONE